# Effect of the size of environment on the steady-state entanglement and coherence via collision model

Xiao-Ming Li, Zhong-Xiao Man, and Yun-Jie Xia[1]

School of Physics and Physical Engineering, Shandong Provincial Key Laboratory of Laser Polarization and Information Technology, Qufu Normal University, 273165, Qufu, China

**Abstract:** The steady-state properties of an open quantum system are investigated via the collision model method of system-reservoir interaction. In our collision model, the system of interest consists of two coupled qubits, each of which interacts with its own independent thermal reservoir. Each thermal reservoir is modeled as a set of clusters of qubits (or linear harmonic oscillators). First, the steady-state entanglement of the system is studied. We show that collective interaction between the system and the elements (qubits or linear harmonic oscillators) in the clusters is beneficial to the generation and enhancement of the steady-state entanglement. And increasing the size of the clusters forming the low-temperature thermal reservoir is more conducive to the improvement of steady-state entanglement. Remarkably, we show that the steady-state entanglement can be greatly improved by choosing the suitable size of the clusters forming the thermal reservoirs. We also study the effect of the size of the cluster on the steady-state coherence. The numerical results show that for the qubit clusters, whether the steady-state coherence of the system can be enhanced by increasing the size of clusters depends on the coupling strength between the two system qubits and the coupling strength between the system and the thermal reservoirs. While for the case of the harmonic oscillator clusters, in addition to the coupling strengths, whether the steady-state coherence can be enhanced also depends on the temperature of the thermal reservoirs.

**Keywords:** collision model, steady-state entanglement, steady-state coherence, master equation

## 1. Introduction

Quantum entanglement and quantum coherence are recognized as essential physical resources in emerging several fields of quantum technology, such as quantum computation and quantum information[1], quantum cryptography[2-4], quantum metrology[5-6], quantum thermodynamics[7-9]. It is widely known that quantum entanglement and quantum coherence must be attached to an actual quantum system. However, all realistic quantum systems cannot be isolated, but would be affected by the external environment. The inevitable interaction between the system and the environment will lead to decoherence, which makes the initial entanglement and coherence of a particular quantum system for quantum tasks easy to be destroyed. This has seriously hindered the application of quantum entanglement and quantum coherence as physical resources in these new fields of quantum technology. Hence, the exploration of new measures for protecting and enhancing quantum entanglement and coherence in different environments or at least delaying their decay is of considerable practical significance.

So far, many methods have been proposed to generate, protect and improve the quantum entanglement[10-24] and quantum coherence[25-30]. Interestingly, it has been found that noise and dissipation can be exploited for the generation of entanglement and coherence. In particular, more recently, the steady-state entanglement[16,20,22,24,31-37] and coherence[38-42] of an open quantum system have attracted much attention. In [32,33], the authors studied the steady-state entanglement of two interacting qubits coupled to different heat baths and found that the temperature gradient

---

[1] Corresponding author. E-mail address: yunjiexia@126.com

can enhance the entanglement under certain conditions. In Ref. [31], a new protocol has been proposed to generate the steady-state entanglement of a bipartite quantum system from incoherent resources. Ref. [22] has found that the steady-state entanglement of a coupled two-qubit can be enhanced by leading into a third thermal reservoir which is common to both qubits. In Ref. [40], the authors have shown that the steady-state coherence can be generated spontaneously in a two-level system in contact with a single thermal bath by the composite system-bath interaction. In Ref. [42], the authors have shown that the steady-state coherence in the two-level system can be enhanced by repeated system-bath interactions. In their model, the effective bath is modeled as a stream of clusters of qubits (or linear harmonic oscillators) in thermal states.

In this paper, focusing on a coupled two-qubit system, we explore the feasibility of enhancing the steady-state entanglement and the steady-state coherence in the open system by collision model[43-53] which can be straightforwardly realized using superconducting quantum circuits[54] and cold trapped ions[55]. Different from the previous collision models, in our model, each thermal reservoir is simulated by a set of qubit (or linear harmonic oscillator) clusters initially prepared in the thermal state. The results show that the steady-state entanglement and the steady-state coherence can be substantially enhanced by increasing the size of the cluster appropriately. Compared with the collision model used to study entanglement in Ref. [37], we extend each element

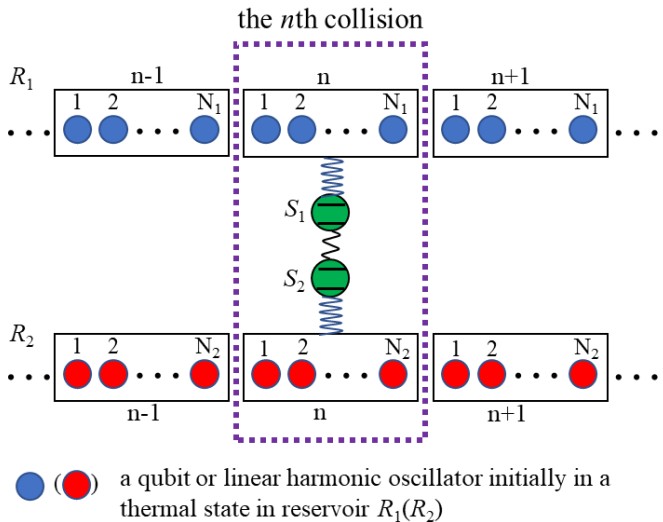

Fig.1. Schematic of collision model considered in this paper. In the $n$th collision, qubit $S_i$ collides with the $n$th cluster in reservoir $R_i$. After collision, the two clusters are discarded, and immediately two system qubits interact with a new cluster in corresponding heat reservoir, respectively. The process is then repeated sequentially.

constituting the heat reservoirs in this model from one qubit to a cluster composed of multiple qubits (or harmonic oscillators). This improvement enables our model to study the effects of various coherences of the bath on the thermalization [49] of the system and local heat flow [56].

## 2. Model and Quantum master equation

In the model we consider, the open system consists of two coupled qubits $S_1$ and $S_2$, with respective frequency $\omega_{S_1}$ and $\omega_{S_2}$. And each qubit $S_i$ ( $i=1,2$ ) interacts with its heat reservoir $R_i$. Here, the heat reservoir $R_i$ is simulated as a collection of clusters of qubits (or linear harmonic oscillators), and each cluster in the heat reservoir $R_i$ is composed of $N_i$

non-correlated and identical qubits (or linear harmonic oscillators) with frequency $\omega_i$, which are initially prepared in the thermal states with the temperature $T_i$ (without loss of generality, we assume $\Delta T = T_2 - T_1 > 0$ in this paper), as depicted in Fig.1. Each system qubit $S_i$ collides collectively with one cluster of $N_i$ non-correlated qubits (or harmonic oscillators) in the reservoir $R_i$ at a time in a short period of time $\tau$. After each collision, the clusters in the two heat reservoirs that have collided with the system qubits are discarded and the system qubit $S_i$ immediately interacts with the next fresh new cluster in the heat reservoir $R_i$. Then the process is repeated sequentially. Here, we assume that the time interval between the two successive collisions is negligible compared to the time $\tau$ and that each cluster in the two heat reservoirs will not share any initial correlations with the system of interest. So, during the time of interaction of duration $\tau$, the total Hamiltonian can be written as

$$\hat{H} = \hat{H}_S + \hat{H}_R + \hat{H}_{SR},\tag{1}$$

where $\hat{H}_S = \frac{1}{2}\omega_{S_1}\hat{\sigma}_{S_1}^z + \frac{1}{2}\omega_{S_2}\hat{\sigma}_{S_2}^z + \xi(\hat{\sigma}_{S_1}^+\hat{\sigma}_{S_2}^- + \hat{\sigma}_{S_1}^-\hat{\sigma}_{S_2}^+)$ is the Hamiltonian of the system of interest, with $\hat{\sigma}_{S_i}^z$ being the Pauli operator, $\hat{\sigma}_{S_i}^+(\hat{\sigma}_{S_i}^-)$ being the raising (lowering) operator for the system qubit $S_i$ ($i=1,2$), and $\xi$ being the coupling strength between the two qubits.

$\hat{H}_R = \sum_{i=1}^{2}\hat{H}_{R_i}$ is the free Hamiltonian of the corresponding two clusters in two heat reservoirs.

And $\hat{H}_{SR} = \sum_{i=1}^{2}\hat{H}_{S_iR_i}$, with $\hat{H}_{S_iR_i}$ denoting the interaction Hamiltonian between the subsystem $S_i$ and the cluster in the corresponding heat reservoir $R_i$.

A quantum master equation describing the evolution of the system is next constructed. For the convenience of taking the continuous time limit, the system-reservoir interaction Hamiltonian is rescaled by an interaction time factor $\frac{1}{\sqrt{\tau}}$ [30,45-47]. Thus, the total Hamiltonian can be reformulated as

$$\hat{H} = \hat{H}_S + \hat{H}_R + \frac{1}{\sqrt{\tau}}\hat{H}_{SR}.\tag{2}$$

We assume that the system qubit $S_i$ ($i=1,2$) collides with the first cluster in the corresponding

heat reservoir $R_i$ at time $t = 0$. Then the state of the total system after the *n*th collision can be

expressed as $\rho(n\tau) = \rho_S(n\tau) \otimes \rho_R^{n+1}$, where $\rho_S(n\tau)$ is the state of the system of interest after

the *n*th collision. $\rho_R^{n+1} = \rho_{R_1}^{n+1} \otimes \rho_{R_2}^{n+1}$ is the total state of the forthcoming two clusters in the two

reservoirs, with $\rho_{R_i}^{n+1}$ being the state of the (*n*+1)th cluster in the heat reservoir $R_i$. The reduced

density matrix of the system of interest after the (*n*+1)th collision is dominated by the map

$$
\begin{aligned}
\rho_S[(n+1)\tau] &= Tr_R\{\hat{U}[\rho_S(n\tau) \otimes \rho_R^{n+1}]\hat{U}^\dagger\} \\
&= Tr_R\{e^{-i\hat{H}\tau}[\rho_S(n\tau) \otimes \rho_R^{n+1}]e^{i\hat{H}\tau}\}
\end{aligned}
\tag{3}
$$

where $\hat{U} = e^{-i\hat{H}\tau}$ is the unitary evolution operator and $Tr_R$ denotes the partial trace over the

heat reservoirs degrees of freedom. We expand the evolution operator $\hat{U}$ up to second order in

$\tau$ and put it into Eq. (3) together with Eq. (2), leads to

$$
\begin{aligned}
&\rho_S[(n+1)\tau] - \rho_S(n\tau) \\
&\quad = -i\tau[\hat{H}_S, \rho_S(n\tau)] - \frac{\tau}{2}Tr_R\{[\hat{H}_{SR},[\hat{H}_{SR}, \rho_S(n\tau) \otimes \rho_R]]\}
\end{aligned}
\tag{4}
$$

In the limit $\tau \to 0$, the master equation describing the evolution of the system is obtained, as
follows

$$
\begin{aligned}
\frac{d\rho_S(t)}{dt} &= \lim_{\tau \to 0} \frac{\rho_S[(n+1)\tau] - \rho_S(n\tau)}{\tau} \\
&= -i[\hat{H}_S, \rho_S] - \frac{1}{2}Tr_R\{[\hat{H}_{SR},[\hat{H}_{SR}, \rho_S \otimes \rho_R]]\}
\end{aligned}
\tag{5}
$$

The interaction between the two qubits and their corresponding clusters can be written as

$$
\hat{H}_{SR} = \sum_{i=1}^{2} g_i(\hat{\sigma}_{S_i}^+ \hat{J}_i^- + \hat{\sigma}_{S_i}^- \hat{J}_i^+),
\tag{6}
$$

where $\hat{J}_i^\pm = \sum_{j=1}^{N_i} \hat{B}_{R_i,j}^\pm$ ($i = 1, 2$) represents the collective raising and lowering operators[31,49] of

each qubit cluster (or collective creation and annihilation operators of each oscillator cluster) in

the reservoir $R_i$. More specifically, $\hat{B}_{R_i,j}^\pm = \hat{\sigma}_{R_i,j}^\pm$ for the cluster of qubits and

$\hat{B}_{R_i,j}^-(\hat{B}_{R_i,j}^+) = \hat{a}_{R_i,j}(\hat{a}_{R_i,j}^\dagger)$ for the cluster of oscillators, with $\hat{\sigma}_{R_i,j}^-$ ($\hat{a}_{R_i,j}$) and $\hat{\sigma}_{R_i,j}^+$ ($\hat{a}_{R_i,j}^\dagger$)

being the individual lowering (annihilation operators) and raising operators (creation operators) in

the reservoir clusters, respectively. $g_i$ is the coupling between the qubit $S_i$ and each cluster in

the reservoir $R_i$. Putting Eq. (6) into Eq. (5), the quantum master equation can be written as

$$\frac{d\rho_S(t)}{dt} = -i[\hat{H}_S, \rho_S] + \sum_{i=1}^{2} \mathcal{L}_i(\rho_S) \quad , \tag{7}$$

where

$$\begin{aligned}
\mathcal{L}_i(\rho_S) &= \gamma_i \left\langle \hat{J}_{R_i}^- \hat{J}_{R_i}^+ \right\rangle_R (\hat{\sigma}_{S_i}^- \rho_S \hat{\sigma}_{S_i}^+ - \frac{1}{2}\{\hat{\sigma}_{S_i}^+ \hat{\sigma}_{S_i}^-, \rho_S\}) \\
&+ \gamma_i \left\langle \hat{J}_{R_i}^+ \hat{J}_{R_i}^- \right\rangle_R (\hat{\sigma}_{S_i}^+ \rho_S \hat{\sigma}_{S_i}^- - \frac{1}{2}\{\hat{\sigma}_{S_i}^- \hat{\sigma}_{S_i}^+, \rho_S\})
\end{aligned} \quad , \tag{8}$$

with $\gamma_i = g_i^2$ being the dissipation rate. For the sake of simplicity, in the following, we assume $\gamma_1 = \gamma_2 = \gamma$. $\left\langle \hat{X} \right\rangle_R = Tr_R(\rho \hat{X})$ represents the average of $\hat{X}$, and $\{\hat{X}, \hat{Y}\} = \hat{X}\hat{Y} + \hat{Y}\hat{X}$ stands for the anti-commutator.

Note that, in this paper, we are concerned only with the steady-state properties of the system. We adopt concurrence[57-58] to measure the steady-state entanglement. According to the master equation (i.e., Eq. (7)), in the ordered basis $\left\{ |ee\rangle_{S_1 S_2}, |eg\rangle_{S_1 S_2}, |ge\rangle_{S_1 S_2}, |gg\rangle_{S_1 S_2} \right\}$, the density matrix of the system of interest in the steady state can be expressed as

$$\rho^{SS} = \begin{pmatrix} \rho_{11}^{SS} & 0 & 0 & \rho_{14}^{SS} \\ 0 & \rho_{22}^{SS} & \rho_{23}^{SS} & 0 \\ 0 & \rho_{32}^{SS} & \rho_{33}^{SS} & 0 \\ \rho_{41}^{SS} & 0 & 0 & \rho_{44}^{SS} \end{pmatrix}. \tag{9}$$

The concurrence is given as

$$C(\rho) = 2 \max\{0, |\rho_{23}^{SS}| - \sqrt{\rho_{11}^{SS} \rho_{44}^{SS}}, |\rho_{14}^{SS}| - \sqrt{\rho_{22}^{SS} \rho_{33}^{SS}}\}. \tag{10}$$

In addition, to quantify the steady-state coherence of the coupled two-qubit system, we use the $l_1$-norm of coherence measure[59], which is expressed as

$$C_{l_1}^{SS} = \sum_{i \neq j} |\rho_{ij}^{SS}|. \tag{11}$$

### 3. Heat reservoirs composed of qubit clusters

This section presents the case in which each cluster in the reservoir $R_i$ is composed of $N_i$ ($i = 1, 2$) non-correlated and identical qubits. Under this circumstance, the state $\rho_{R_i}^n$ of the $n$th cluster in the reservoir $R_i$ is given by the tensor product of the states of $N_i$ qubits, i.e.,

$\rho_{R_i}^n = \otimes_{j=1}^{N_i} \eta_{R_i}^j$, where $\eta_{R_i}^j = exp(-\frac{\omega_i}{2T_i} \hat{\sigma}_{R_i,j}^z) / Z_{R_i,j}$ $(k_B = 1, \hbar = 1)$ is the initial state of the $j$th

qubit forming each cluster in the reservoir $R_i$, with $Z_{R_i,j} = TrExp(-\frac{\omega_i}{2T_i}\hat{\sigma}_{R_i,j}^z)$ being the partition function. Therefore, the expectation values $\langle \hat{J}_i^- \hat{J}_i^+ \rangle_R$ and $\langle \hat{J}_i^+ \hat{J}_i^- \rangle_R$ in Eq. (7) can be easily obtained, as follows

$$\langle \hat{J}_i^- \hat{J}_i^+ \rangle_R = \frac{N_i}{2}(1 + \tanh\frac{\omega_i}{2T_i}), \tag{12}$$

$$\langle \hat{J}_i^+ \hat{J}_i^- \rangle_R = \frac{N_i}{2}(1 - \tanh\frac{\omega_i}{2T_i}). \tag{13}$$

By substituting Eq. (12) and Eq. (13) into Eq. (7), and letting $\frac{d\rho_S(t)}{dt} = 0$, the steady-state solutions can be obtained. For simplicity, we assume $\omega_{S_1} = \omega_{S_2} = \omega_1 = \omega_2 = \omega$ in this section.

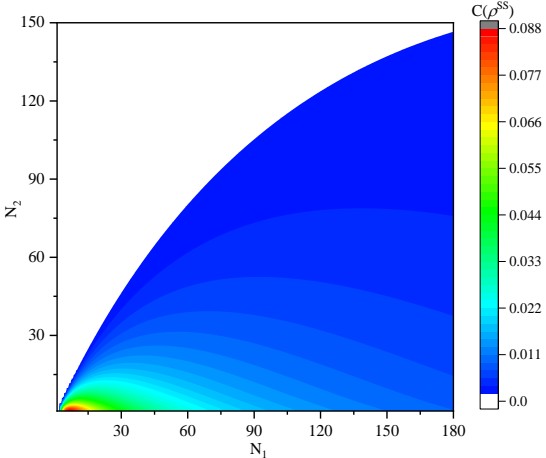

Fig.2. The contour plot of the steady-state concurrence $C(\rho^{SS})$ for different $N_1$ and $N_2$. The parameters

are $T_1 = 0.01\omega$, $\Delta T = 5\omega$, $\xi = 0.1\omega$ and $\gamma = 0.1\omega$.

Firstly, we discuss the steady-state entanglement of the system. In Fig.2, we show an overall picture for the dependence of the steady-state concurrence $C(\rho^{SS})$ on the size of each cluster in the two reservoirs (i.e., $N_1$ and $N_2$) for a given temperature of two heat reservoirs. We can see that, in terms of improving the steady-state entanglement of the system, it is better to increase the number of qubits in the cluster forming the low-temperature heat reservoir separately than to increase the size of clusters in both heat reservoirs simultaneously, which is beyond our expectations. What's more, the steady-state concurrence decreases monotonically with the number $N_2$ of qubits. For a given $N_1$, when $N_2$ increases to a certain value, the steady-state

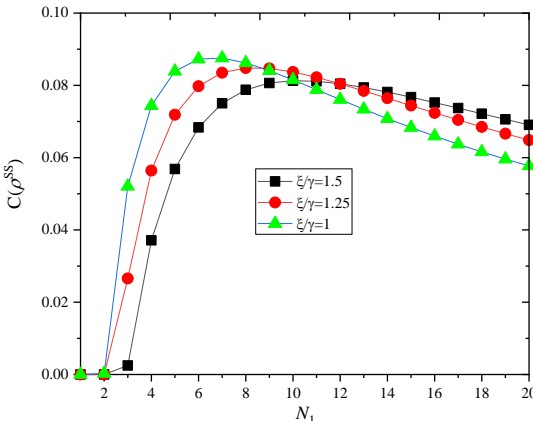

Fig.3. The steady-state concurrence $C(\rho^{ss})$ as a function of $N_1$ for different values of $\xi/\gamma$. The

parameters are $T_1 = 0.01\omega$, $\Delta T = 5\omega$ and $N_2 = 1$.

entanglement vanishes completely. For a given $N_2$, the steady-state concurrence first increases

and then decreases with the increase of $N_1$. Therefore, there must be a critical value $N_1^{QC}$ for

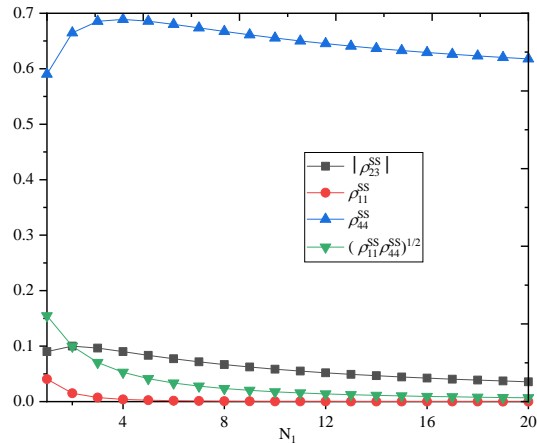

Fig.4. $\left|\rho_{23}^{SS}\right|$, $\rho_{11}^{SS}$, $\rho_{44}^{SS}$ and $\sqrt{\rho_{11}^{SS}\rho_{44}^{SS}}$ as a function of $N_1$. The parameters are $\xi = 0.1\omega$,

$\gamma = 0.1\omega$, $T_1 = 0.01\omega$, $\Delta T = 5\omega$ and $N_2 = 1$.

the size of the clusters constituting the low-temperature thermal reservoir, in which case the steady-state concurrence of the system reaches the maximum. In other words, in order to obtain greater entanglement, the size of the clusters constituting the low-temperature thermal reservoir should be $N_1^{QC}$. This shows that it is feasible to enhance the steady-state entanglement by the collective interaction between the system of interest and the cluster formed by more than one independent and identical qubits. Note that $N_1^{QC}$ depends on the ratio $\xi/\gamma$. According to the

numerical calculation results, we find that $N_1^{QC} \approx 7\dfrac{\xi}{\gamma}$ when $N_2 = 1$, as shown in Fig.3. In

order to understand these observations more deeply, we study the effects of $N_1$ on $\left|\rho_{23}^{SS}\right|$ and

populations of the system which are involved in the steady-state concurrence, as shown in Fig.4.

The increase of $N_1$ causes $\left|\rho_{23}^{SS}\right|$ and the population $\rho_{44}^{SS}$ to increase at first and then decrease,

and leads to the population $\rho_{11}^{SS}$ rapid attenuation. It can be seen that when the value of $N_1$ is

small, $\left|\rho_{23}^{SS}\right| < \sqrt{\rho_{11}^{SS}\rho_{44}^{SS}}$, so the steady-state concurrence $C(\rho^{SS}) = 0$. When $N_1$ is large, the

population $\rho_{11}^{SS} \to 0$, so $C(\rho^{SS}) \to \left|\rho_{23}^{SS}\right|$. When $N_1 \to \infty$, since $\left|\rho_{23}^{SS}\right| \to 0$, the steady-state

entanglement vanishes (i.e., $C(\rho^{SS}) \to 0$). As a consequence, the variation of the increment of

the steady-state concurrence exhibits nonlinear relation with $N_1$.

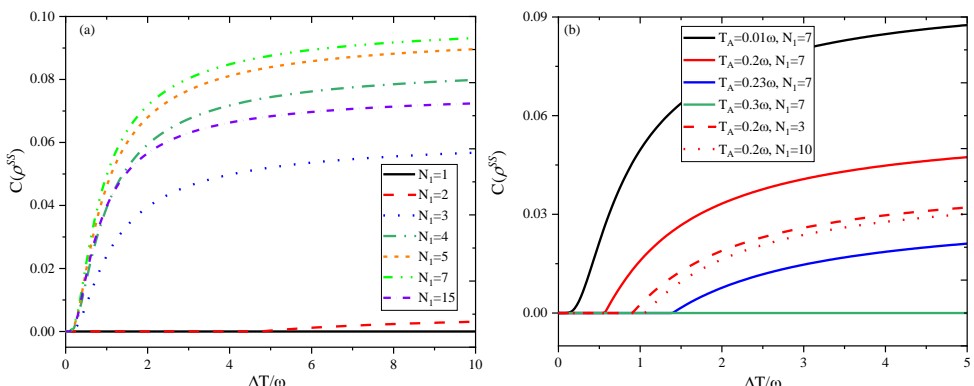

Fig.5 (a) $C(\rho^{SS})$ as a function of $\Delta T$ for different $N_1$ with $T_1 = 0.01\omega$; (b) $C(\rho^{SS})$ as a function of

$\Delta T$ for different $T_1$。 The parameters are $N_2 = 1$, $\xi = 0.1\omega$, $\gamma = 0.1\omega$。

To fully grasp the effect of the size of the clusters in the heat reservoirs on steady-state

entanglement, in Fig. 5(a), we plot the steady-state concurrence $C(\rho^{SS})$ as a function of the

temperature difference $\Delta T$ for different $N_1$ when $N_2 = 1$. For a given $\Delta T$, with the

chosen parameters, when $N_1 = 7$, $C(\rho^{SS})$ reaches a maximum (namely, $N_1^{QC} = 7$), as shown

by the green dotted line. This indicates that the critical size $N_1^{QC}$ is independent of the reservoir

temperature. For a given $N_1 > 1$, $C(\rho^{SS})$ increases with the increase of $\Delta T$, and when

$N_1 = N_1^{QC}$, $C(\rho^{SS})$ increases fastest with $\Delta T$. In addition, we find that when $N_1 = 1$, as

shown with the black solid line in Fig.5, $C(\rho^{SS}) = 0$, which is consistent with the conclusion in

Ref. [37]. However, when $N_1 \geq 2$, the steady-state entanglement arises. This indicates that

increasing the number of qubits forming the clusters in the reservoir with a lower temperature can

assist the generation of the steady-state entanglement. In Fig.5(b), we show the steady-state

concurrence $C(\rho^{SS})$ as a function of the temperature difference $\Delta T$ for different $T_1$.

Obviously, for a given $N_1$ (e.g., $N_1 = 7$, see the solid line), $C(\rho^{SS})$ decreases with the increase

of $T_1$. Only when the temperature of the low-temperature heat reservoir is relatively low and the

temperature difference is large enough, can the steady-state entanglement occur, which is

consistent with the conclusion obtained in Ref. [20, 35]. Moreover, by comparing the three red

curves in Fig.5(b), it can be found that when $N_1 = N_1^{QC}$, the temperature difference $\Delta T$

required to generate steady-state entanglement is the smallest.

Next, we discuss the steady-state coherence of the system in this cluster scenario. the

steady-state coherence measured by the $l_1$-norm can be expressed as

$$C_{l_1}^{SS} = \frac{2N_1 N_2 \xi \gamma}{(N_1 + N_2)(N_1 N_2 \gamma^2 + 4\xi^2)}[\tanh \frac{\omega}{2T} - \tanh \frac{\omega}{2(T + \Delta T)}]. \tag{14}$$

Here, we have re-labeled the temperatures of the two heat reservoirs as $T$ and $T + \Delta T$,

respectively. We can see that for any given coupling strength, the sizes of the clusters that

maximize $C_{l_1}^{SS}$ should meet $N_1 = N_2$. Therefore, for the case of $N_1 = N_2 = N$, the analytical

expression of the steady-state coherence can be reduced to

$$C_{l_1}^{SS} = \frac{N \xi \gamma}{N^2 \gamma^2 + 4\xi^2} A(T, \Delta T), \tag{15}$$

where $A(T, \Delta T) = \tanh \frac{\omega}{2T} - \tanh \frac{\omega}{2(T + \Delta T)}$. One can find that when $N = 2\frac{\xi}{\gamma}$, the

steady-state coherence $C_{l_1}^{SS}$ reaches its maximum. That is to say, in order to enhance the

steady-state coherence of the system of interest as much as possible, the chosen value of $N$

should be roughly $2\frac{\xi}{\gamma}$. It should be noted that if $2\frac{\xi}{\gamma} \leq 1$, increasing the size of the cluster can

only weaken the steady-state coherence in the system. In order to display the above conclusions

more intuitively, the numerical results when $\dfrac{\xi}{\gamma}=1$ are given in Fig. 6. It can be seen that by increasing the size of the cluster beyond $N=1$, e.g., $N=2$ and $3$, the steady-state coherence is enhanced, and reaches the maximum when $N=2$. However, when $N>3$, $C_{l_1}^{SS}$ decreases monotonically with $N$. According to Eq. (15), when $N \gg 2\dfrac{\xi}{\gamma}$, the steady-state coherence $C_{l_1}^{SS}$ is approximately proportional to $\dfrac{1}{N}$. This explains why the steady-state coherence decreases with $N$ when $N$ is large in Fig. 6.

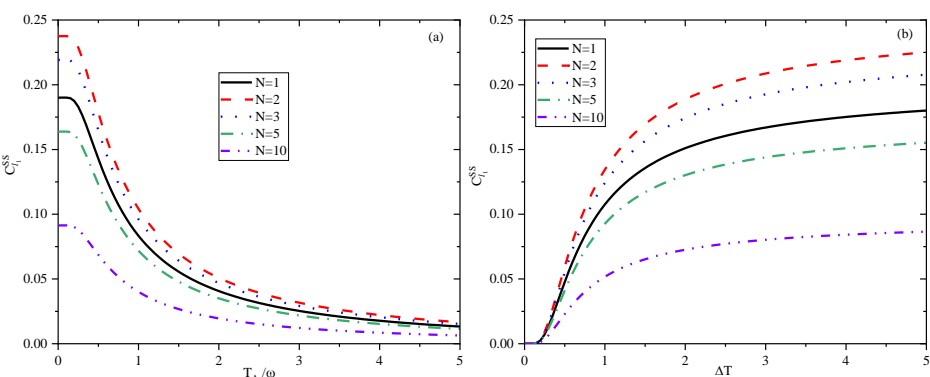

Fig.6. (a) the dependence of the steady-state coherence $C_{l_1}^{SS}$ on $T_1$ with $\Delta T=10\omega$ for different $N$. (b) the dependence of the steady-state coherence $C_{l_1}^{SS}$ on $\Delta T$ with $T_1=0$ for different $N$. The other parameters are $\xi=0.1\omega$ and $\gamma=0.1\omega$.

On the other hand, the effects of temperature $T$ and temperature difference $\Delta T$ on steady-state coherence depend on the function $A(T,\Delta T)$ in Eq. (15). When $\Delta T=0$, obviously, $A(T,\Delta T)=0$, as a result, the steady-state coherence and the steady-state entanglement vanish completely, regardless of the size of the clusters. In other words, if the two heat reservoirs have the same temperature, the steady-state coherence and entanglement will not occur in the two-coupled qubit system. For $T \gg 0$, we approximate $A(T,\Delta T) \rightarrow \omega\dfrac{\Delta T}{2T^2}$. Therefore, for a given $\Delta T$, the steady-state coherence decays in accordance with the inverse-square law of temperature in the high temperature limit until it disappears completely. This can be seen clearly from the numerical results plotted in Fig.6(a). Moreover, for a given $N>1$, compared with $N=1$, the lower the temperature $T$, as shown in Fig.6(a), (or the greater the temperature difference $\Delta T$, as shown in Fig.6(b)), the larger the increment of $C_{l_1}^{SS}$.

## 4. Heat reservoirs composed of oscillator clusters

In this section, we assume that each cluster in the heat reservoir $R_i$ ($i = 1, 2$) is composed

of $N_i$ independent and identical linear harmonic oscillators in the thermal state. For each linear

harmonic oscillator forming the clusters in the reservoir $R_i$, there must be

$$\left\langle \hat{a}^\dagger_{R_i,j} \hat{a}_{R_i,j} \right\rangle_R = \bar{n}_{R_i}, \qquad \left\langle \hat{a}_{R_i,j} \hat{a}^\dagger_{R_i,j} \right\rangle_R = \bar{n}_{R_i} + 1, \tag{16}$$

where $\bar{n}_{R_i} = (e^{\omega_i/T_i} - 1)^{-1}$ is the average occupation number. Thus, in this case, the explicit form

of the expectation values $\left\langle \hat{J}^-_i \hat{J}^+_i \right\rangle_R$ and $\left\langle \hat{J}^+_i \hat{J}^-_i \right\rangle_R$ in Eq. (7) can be easily derived as follows

$$\left\langle \hat{J}^-_i \hat{J}^+_i \right\rangle_R = \frac{N_i}{2}(\coth\frac{\omega_i}{2T_i} + 1), \tag{17}$$

$$\left\langle \hat{J}^+_i \hat{J}^-_i \right\rangle_R = \frac{N_i}{2}(\coth\frac{\omega_i}{2T_i} - 1). \tag{18}$$

Analogously to the previous section, we can obtain the steady-state properties of the system of interest by substituting these results into Eq. (7).

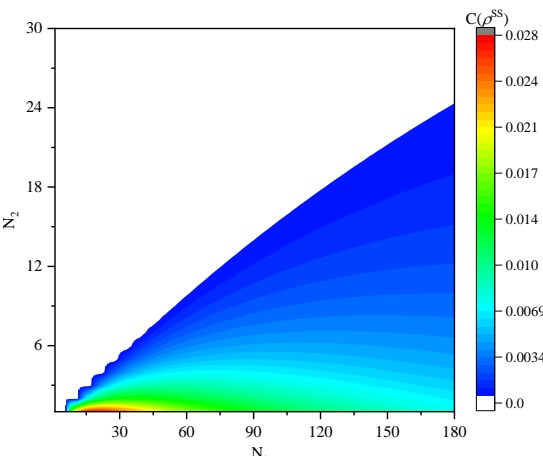

Fig.7. The contour plot of $C(\rho^{SS})$ for different $N_1$ and $N_2$. The parameters are $T_1 = 0.01\omega$,

$\Delta T = 5\omega$, $\xi = 0.1\omega$ and $\gamma = 0.1\omega$.

First, we investigate the effect of the size of the cluster on the steady-state entanglement. In

Fig. 7, we plot the steady-state concurrence $C(\rho^{SS})$ as a function of the size of the clusters in

the two reservoirs. Compared with the case of qubit clusters discussed in the previous section, here the behavior of the steady-state concurrence exhibits certain variations although the overall trends in both cases have similarities (see Fig. 2 and Fig. 7, respectively). It can be seen that the maximum value of the steady-state concurrence shown in Fig. 7 is far less than that shown in Fig. 2. This indicates that to obtain the larger steady-state entanglement, we should select the thermal reservoirs composed of qubit clusters. We can also notice that, in Fig. 7, compared with the case of

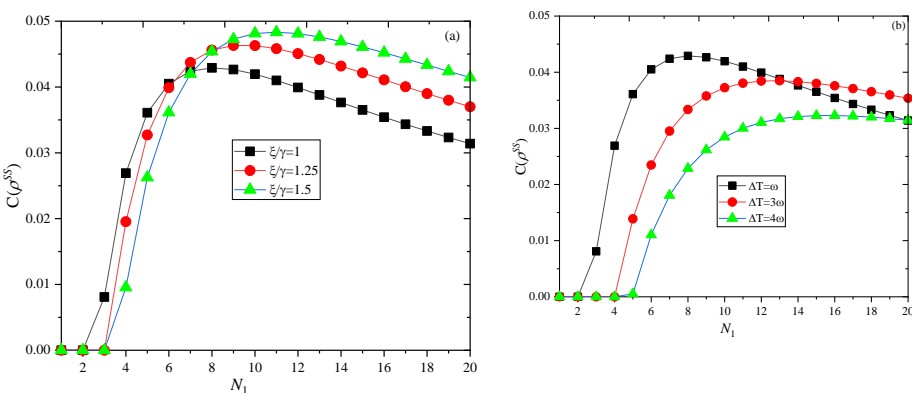

Fig.8. (a) $C(\rho^{SS})$ as a function of $N_1$ for different $\xi/\gamma$ with $\Delta T = \omega$. (b) $C(\rho^{SS})$ as a function

of $N_1$ for different $\Delta T$ with $\xi/\gamma = 1$. The parameters are $T_1 = 0.01\omega$ and $N_2 = 1$.

the qubit clusters (see Fig. 2), the steady-state concurrence decreases faster with the increase of $N_2$. If $N_1 \leq N_2$, the steady-state entanglement does not arise. Indeed, the steady-state entanglement in the system appears only if the value of $N_1$ is several times larger than the value of $N_2$, as shown in Fig.7. Another difference between the two is that, for the case of harmonic oscillator clusters, the critical dimension $N_1^{OC}$ for maximizing the steady-state concurrence also depends on the reservoir temperature. With the chosen parameters, we approximate $N_1^{OC} \approx 8\frac{\xi}{\gamma}\beta(\Delta T)$, as shown in Fig. 8, where $\beta(\Delta T)$ reflects the effect of reservoir temperature on $N_1^{OC}$. One can see from Fig. 8(b), the critical size $N_1^{OC}$ increases with the increase of $\Delta T$.

In Fig.9, we show the steady-state concurrence $C(\rho^{SS})$ as a function of the temperature difference $\Delta T$ for different $N_1$. For a small value of $N_1$, e.g., $N_1 \leq 2$, no matter what the temperature difference is, the steady-state entanglement in the system of interest does not arise. For a given $N_1 \geq 3$, with the increase of temperature difference, the steady-state entanglement arises and its variation exhibits the behavior of increasing first and then decreasing and vanishes eventually. This indicates that, similar to the case of the qubit clusters, increasing the number of harmonic oscillators in each cluster also contributes to the generation of steady-state entanglement. Additionally, although the variations of the maximum of $C(\rho^{SS})$ with the number $N_1$ are nonmonotonic, it is obvious that the steady-state entanglement can be greatly enhanced in a certain region of $\Delta T$ by increasing the size of each cluster. It is important to mention that the

larger the value of $N_1$, the larger the interval of the temperature difference $\Delta T$ where the $C(\rho^{ss})$ maintains non-zero. That is, increasing the size of the harmonic oscillator cluster in the low-temperature reservoir can lower the decay rate of the steady-state entanglement and prolong the nonzero concurrence to a larger region of temperature difference. This has a certain

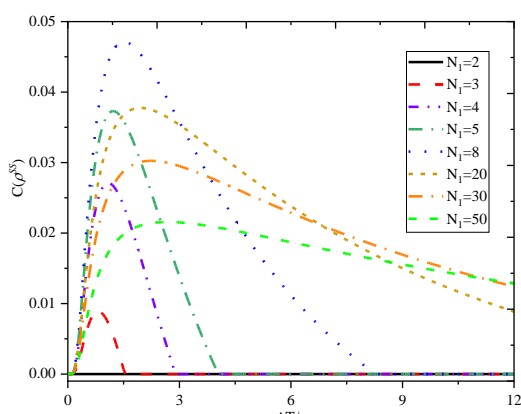

Fig.9. $C(\rho^{ss})$ as a function of $\Delta T$ for different $N_1$. The parameters are $T_1 = 0.01\omega$, $N_2 = 1$, $\xi = 0.1\omega$ and $\gamma = 0.1\omega$.

reference value for the protection and practical application of steady-state entanglement.

In what follows, we briefly discuss the effect of the size of the cluster on the steady-state coherence in this case. The $l_1$-norm of the steady-state coherence can be formulated as

$$C_{l_1}^{SS} = \frac{N_1 N_2 \xi \gamma [\coth \frac{\omega}{2(T+\Delta T)} - \coth \frac{\omega}{2T}]}{[N_1 \coth \frac{\omega}{2T} + N_2 \coth \frac{\omega}{2(T+\Delta T)}][4\xi^2 + N_1 N_2 \gamma^2 \coth \frac{\omega}{2T} \coth \frac{\omega}{2(T+\Delta T)}]}. \quad (19)$$

In the special case of $N_1 = N_2 = N$, the above equation can be simplified as

$$C_{l_1}^{SS} = \frac{V(T,\Delta T)}{\frac{4}{N}\frac{\xi}{\gamma} + N\frac{\gamma}{\xi}W(T,\Delta T)}, \quad (20)$$

where

$$V(T,\Delta T) = \frac{\coth \frac{\omega}{2(T+\Delta T)} - \coth \frac{\omega}{2T}}{\coth \frac{\omega}{2(T+\Delta T)} + \coth \frac{\omega}{2T}}, \quad (21)$$

and

$$W(T,\Delta T) = \coth \frac{\omega}{2T} \coth \frac{\omega}{2(T+\Delta T)}. \quad (22)$$

Consequently, we can easily obtain the specific value of $N$ which maximizes the steady-state coherence

$$N_{sp} = \frac{2}{\sqrt{W(T, \Delta T)}} \frac{\xi}{\gamma}. \tag{23}$$

It can be found according to Eq. (21) that when $\Delta T = 0$, the steady-state coherence completely vanishes, which is similar to the conclusion in the previous section. For a given $\Delta T > 0$, the condition $N_{sp} \geq 2$ has to be satisfied to enhance the steady-state coherence by increasing the size of the cluster. What needs to be stressed is that, different from the previous section, the value of $N_{sp}$ depends on temperature ($T$) and temperature difference ($\Delta T$) in addition to

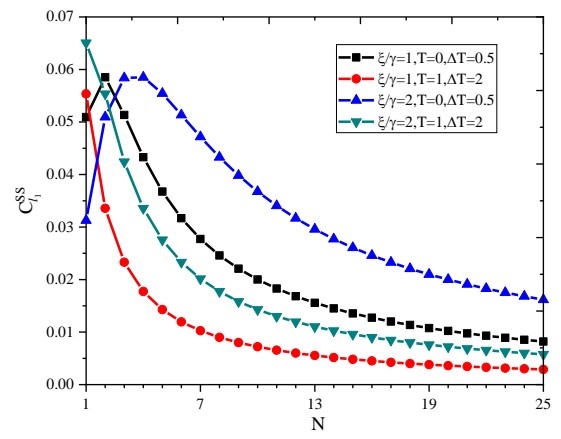

Fig.10. the steady-state coherence $C_{l_1}^{SS}$ as a function of $N$ for different $\xi / \gamma$, $T$ and $\Delta T$.

coupling strengths $\xi$ and $\gamma$. Therefore, according to Eqs. (22) and (23), at relatively low temperature and small temperature difference, the steady-state coherence can be improved by changing the size of the clusters constituting the heat reservoirs. This observation can be visualized in Fig.10. However, in the high temperature limit, we can definitely deduce $W(T, \Delta T) \gg 1$, which lead to $N_{sp} < 1$. That is to say, in the high temperature limit, it is impossible to enhance the steady-state coherence in the system by increasing the number of linear harmonic oscillators forming each cluster.

## 5. Conclusions

In conclusion, we have studied the steady-state properties of two coupled qubits based on the collision model. In our collision model, each reservoir coupled to the system of interest is modeled as a set of clusters of qubits (or linear harmonic oscillators), and each qubit (or linear harmonic oscillator) forming the clusters is initially prepared in a thermal state. Each system qubit interacts collectively with a cluster in the corresponding heat reservoir at a time. Then the clusters in the two reservoirs that have collided with the system qubits are discarded, and immediately the next round of collision occurs.

We have investigated the steady-state entanglement between the two coupled qubits. For the two cases we considered, we find that increasing the size of the clusters forming the low-temperature heat reservoir alone is more conducive to the improvement of the steady-state entanglement of the two coupled qubits. In particular, we show that the steady-state entanglement of the two qubits can be enhanced to the most extent by choosing the appropriate size of each cluster for the given temperatures of the two heat reservoirs. Previous studies [37] have shown that if the intracollisions are not considered, the steady-state entanglement will not arise when there is only one qubit in each cluster. By contrast, our results show that the collective collision can induce

steady-state entanglement even if intracollisions is not considered. This is exactly the embodiment of the advantage of the collision model we considered. Moreover, we also find that increasing the size of each cluster in the high-temperature reservoir separately can only reduce the steady-state entanglement in the system of interest. Comparing the two cases, under the same conditions, the number of elements forming the cluster is different when the steady-state concurrence increases to the maximal value, and the maximum value of the steady-state concurrence that the system can achieve is higher for the qubits-composed baths. This provides guidance for us to choose a reasonable experimental scheme to obtain greater steady-state entanglement.

We have also studied the steady-state coherence in the system. We have provided explicit expressions of the steady-state coherence for these two cases. The conditions for enhancing the steady-state coherence by increasing the number of elements forming clusters have been given. When the clusters are composed of qubits, as long as the interaction strength in the system of interest and the coupling strength between the system and the reservoirs are properly selected, it is possible to increase the steady-state coherence by increasing the size of the cluster for any given heat reservoir temperature $T$ and temperature difference $\Delta T$ (except $\Delta T = 0$). While for the linear harmonic oscillator clusters, the steady-state coherence can be improved by increasing the size of the cluster only at relatively low temperatures. Furthermore, the harmonic oscillators forming the clusters in the thermal reservoir can carry more coherence than qubits. Therefore, intuitively, choosing harmonic oscillator clusters might be more conducive to the improvement of steady-state coherence. However, the results tell us that this is not the case. Especially at a high temperature, increasing the number of harmonic oscillators forming clusters will reduce the steady-state coherence. This considerably differs from the case of a single-qubit system [42].

In practical application, the collision model can be easily implemented using superconducting quantum circuits and linear optical scheme. Hence, our work provides an effective way to enhance the steady-steady entanglement and coherence, which will be helpful to the development of quantum information processing.

**Acknowledgments**

This work is supported by National Natural Science Foundation (China) under Grant No. 61675115 and No.11974209, Taishan Scholar Project of Shandong Province (China) under Grant No. tsqn201812059, and the Youth Technological Innovation Support Program of Shandong Provincial Colleges and Universities under Grant No. 2019KJJ015.

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
