# Peer review of "Effect of the size of environment on the steady-state entanglement and coherence via collision model"

_SciPost Physics_

## Round 1 · Referee Report · Anonymous (Referee 1) · 2022-9-18

Strengths

  1. Simple quantum collisional model, where one can naturally study the stabilization of entanglement via the coupling to two thermal reservoirs.

  2. Transparent results which are easy to be grasped by the reader.

Weaknesses

  1. The amount of entanglement generated in the two-qubit system by this specific collision model seems relatively low.

  2. There is no hint on how the situation may change or improve by considering more sophisticated systems or interacting reservoirs.

  3. No discussion on possible experimental issues nor on the parameter values to be used in a realistic setup.

Report

This manuscript studies a quantum collision model made of a system with two coupled qubits, each of them interacting with a separate heat reservoir. Such kind of bath (for each qubit) is modeled as a collection of identical and independent qubits or of harmonic oscillators, which are supposed to be in a thermal state at finite temperature.
The Authors assume that each qubit in the system collides collectively with a single cluster of uncorrelated qubits (or harmonic oscillators) in a very short period of time. After each collision, the clusters that have collided are discarded and a new collision occurs with another cluster. The process is iterated many times, until a steady-state configuration is reached.
Under these circumstances, they are able to derive a simple master equation for the time evolution of the density matrix of the two-qubit system, which is then used to easily calculate the concurrence and the quantum coherence through the l1-norm of the state.

They report a number of findings on the behavior of such coherence measures on the steady-state, with respect to the number of clusters forming the two reservoirs, their temperature and temperature difference. For example, they find that the entanglement can be generated and improved by increasing the size of clusters composing the low-temperature reservoir, provided its temperature is sufficiently low and the temperature difference with the hot reservoir is large enough. Moreover, the baths of qubits are generally more efficient that those of harmonic oscillators in stabilizing entanglement, although a more complicated dependence on the various bath parameters seem to emerge in the latter case.

The manuscript is generally well written in its parts and understandable, while the results contained are reasonable and I think they may deserve publication somewhere. Yet I am not convinced that the Authors are presenting so striking results to meet the acceptance criteria for SciPost Physics. On the other hand, the properties of the system under investigation can be extracted by a direct solution of a simple two-qubit Lindblad maser equation, which is not surprising to me.

My suggestion is to transfer this manuscript for consideration in the SciPost Physics Core section.

Requested changes

I have however a few comments and questions the Authors should deal with.

  1. In the introduction, it is stated that collision models can be straightforwardly realized using superconducting circuits and trapped ions. It is not obvious to me that this specific model is so easy to implement experimentally. May the Authors comment on this point and on the experimental constraints one may have on the geometry of the system and on the choice of the various parameters?

  2. The Hamiltonian for the two-qubit system contains an isotropic XY interaction plus two transverse-field terms. The system-bath interaction is also of the XY-type (with the system qubit coupled to each single qubit (or boson) in the corresponding bath. Is there a reason (apart from computational reasons) for choosing this specific setup? For example, how would the scenario change by adding a ZZ-interaction term between the qubits or a term which does not conserve the number of excitations in the z direction?

  3. Concerning the construction of the master equation, as far as I understand, one crucial requirement is that the collision time \tau is extremely small to ensure that the bath does not evolve in time. Is this correct? Why is it required to rescale the interaction time by a 1/\sqrt{\tau} factor?

  4. The Authors assume that the dissipation rates for the two qubits are the same. The frequencies of the qubits, as well as of the harmonic oscillators, are also the same. Would the performance of this entanglement generation scheme improve by changing them?

  5. Did the Authors analyze the specific dependence of the concurrence with \Delta T (e.g. in Fig. 5)? Is this logarithmic, or should one expect some plateau emerging for \Delta T -> \infty?

  6. Is there an explanation of why the concurrence seems to be generally smaller for baths of harmonic oscillators, compared to baths of qubits?

  7. Collision models have been recently objects of intense investigation in the quantum scenario. To complement their references, the Authors may find useful to have a look at this recent review paper by F. Ciccarello et al., Phys. Rep. 954, 1 (2022).

---

## Round 1 · Referee Report · Anonymous (Referee 2) · 2022-9-26

Report

The authors investigate the steady-state properties of an open quantum system made of two interacting qubits, each coupled to a series of identical thermal reservoirs (modelled either as a chain of qubits or either as harmonic oscillators) for a time $\tau$. The dynamics is described by a collision model, that is, after each time window $\tau$, the two qubits are suddenly let to interact with the next set of reservoirs for a time $\tau$ and so on. The evolution is described via a Lindblad master equation with standard up and down polarisers for the two qubits.
For both types of environments, the focus is on the dependence on the sizes and temperatures of the clusters in the steady-state entanglement and the steady-state coherence. The latter are both obtained from the numerical calculation of the steady-state reduced density matrix $\rho^{SS}$ via the concurrence (Eq.(10)) and the $l_1$-norm (Eq.(11)).

The main goal of this work is to understand which type of reservoir (and of which size and temperature) leads to a maximisation of steady-state entanglement and coherence. Some conclusions are drawn based on the numerical analysis.

The paper looks correct and contains results that might be of some interest to the community working in related fields. However, in my opinion, the results obtained by the authors are not enough groundbreaking to meet the acceptance criteria of SciPost Physics. I suggest considering the resubmission of the manuscript in SciPost Physics Core after a revision according to the points listed below.

Requested changes

-For a given Hilbert space basis, the steady-state problem can be formulated in terms of the search of a zero-eigenstate for a 16x16 matrix, which is the vectorized form of the Liouvillian in Eq.(8). Given the simple structure of (8), I am wondering if this approach can be useful to complement the numerical study of $\rho^{SS}$ with some analytical calculation.

-Sec.4, page 6, below Eq.(13), the authors write "For simplicity, we assume $\omega_{S_1}=\omega_{S_2}=\omega_1=\omega_2=\omega$ in this section." However, even in Sec.4, the analysis is made only in terms of a parameter $\omega$, which I assumed to have the same meaning as Sec.3 (although it is not specified in the text).

-For both types of environments (Eq.(14) and Eq.(19)), the authors provide an analytical expression for $C^{SS}_{l_1}$. I kindly ask the authors to explain how they obtain these results.

-in the abstract, it is not clear why one should look for an enhancement of the steady-state entanglement. Indeed, in other contexts, one may want to keep the entanglement small. The authors might consider specifying in the abstract that this study is motivated by possible applications with quantum technologies.

-The authors may consider improving the grammar and the formatting of the manuscript. For instance: references have different styles, there is no page numbering etc. Despite not being scientifically relevant, this will help the readability of the manuscript during the peer-review.

---

## Editorial Decision

awaiting_resubmission